# A phenome-wide association study of 26 mendelian genes reveals phenotypic expressivity of common and rare variants within the general population

Catherine Tcheandjieu[1,2,3], Matthew Aguirre[2,3,4], Stefan Gustafsson[1,3], Priyanka Saha[1,2,3], Praneetha Potiny[1,2,3], Melissa Haendel[5], Erik Ingelsson[1,3,6,7], Manuel A. Rivas[4,3], James R. Priest[1,2,3,8]*

1 Stanford Cardiovascular Institute, Stanford University, Stanford, Stanford, California, United States of America, 2 Department of Pediatric Cardiology Stanford University School of Medicine, Stanford, California, United States of America, 3 Department of Medicine, Division of Cardiovascular Medicine, Stanford University School of Medicine, Stanford, California, United States of America, 4 Department of Biomedical Data Science, Stanford University School of Medicine, Stanford, California, United States of America, 5 Department of Medical Informatics and Clinical Epidemiology, School of Medicine, Oregon Health & Science University (OHSU), Oregon, United States of America, 6 Department of Medicine, Division of Cardiovascular Medicine, Stanford University School of Medicine, Stanford, California, United States of America, 7 Stanford Diabetes Research Center, Stanford University, Stanford, California, United States of America, 8 Chan-Zuckerberg Biohub, San Francisco, California, United States of America

* jpriest@stanford.edu

**Data Availability Statement:** All results described in the current manuscript are available in the supporting information submitted along with the

## Abstract

The clinical evaluation of a genetic syndrome relies upon recognition of a characteristic pattern of signs or symptoms to guide targeted genetic testing for confirmation of the diagnosis. However, individuals displaying a single phenotype of a complex syndrome may not meet criteria for clinical diagnosis or genetic testing. Here, we present a phenome-wide association study (PheWAS) approach to systematically explore the phenotypic expressivity of common and rare alleles in genes associated with four well-described syndromic diseases (Alagille (AS), Marfan (MS), DiGeorge (DS), and Noonan (NS) syndromes) in the general population.

Using human phenotype ontology (HPO) terms, we systematically mapped 60 phenotypes related to AS, MS, DS and NS in 337,198 unrelated white British from the UK Biobank (UKBB) based on their hospital admission records, self-administrated questionnaires, and physiological measurements. We performed logistic regression adjusting for age, sex, and the first 5 genetic principal components, for each phenotype and each variant in the target genes (*JAG1*, *NOTCH2 FBN1*, *PTPN1 and RAS-opathy genes*, and genes in the 22q11.2 locus) and performed a gene burden test.

Overall, we observed multiple phenotype-genotype correlations, such as the association between variation in *JAG1*, *FBN1*, *PTPN11* and *SOS2* with diastolic and systolic blood pressure; and pleiotropy among multiple variants in syndromic genes. For example, rs11066309 in *PTPN11* was significantly associated with a lower body mass index, an increased risk of hypothyroidism and a smaller size for gestational age, all in concordance with NS-related

manuscript. The UK Biobank individual level data can be accessed by submitting an application through the following link: https://www.ukbiobank. ac.uk/register-apply/. Scripts are available at d have shared the scripts for analysis at https://github. com/cathynes/PHEWAS-SD.

**Funding:** This project was funded by grants from the National Institutes of Health (R00HL130523 to Dr Priest) and a Stanford CVI-MCHRI Seed Grant to Dr. Priest & Dr. Tcheandjieu. This research has been conducted using the UK Biobank Resource under Application Numbers 24983 (Dr Rivas), 15860 (Dr Priest), and 13721 (Dr Ingelsson). M.A. R. is supported by Stanford University and a National Institute of Health grants (5U01 HG009080 and R01HG010140). M.H. is supported by grants from the National Institute of Health (R24OD011883). The funders had no role in study design, data collection and analysis, decision to publish, or preparation of the manuscript.

**Competing interests:** The authors have declared that no competing interests exist.

phenotypes. Similarly, rs589668 in *FBN1* was associated with an increase in body height and blood pressure, and a reduced body fat percentage as observed in Marfan syndrome.

Our findings suggest that the spectrum of associations of common and rare variants in genes involved in syndromic diseases can be extended to individual phenotypes within the general population.

## Author summary

Standard medical evaluation of genetic syndromes relies upon recognizing a characteristic pattern of signs or symptoms to guide targeted genetic testing for confirmation of the diagnosis. This may lead to missing diagnoses in patients with silent or a low expressed form of the syndrome. Here we take advantage of a rich electronic health record, various phenotypic measurements, and genetic information in 337,198 unrelated white British from the UKBB, to study the relation between single syndromic disease phenotypes and genes related to syndromic disease. We show multiple phenotype-genotype associations in concordance with phenotypes variations found in syndromic diseases. For example, we show that a commonly found variant in *FBN1* was associated with high standing/sitting height ratio and reduced body fat percentage as observed in individuals with Marfan syndrome. Our findings suggest that common and rare alleles in syndromic disease genes are causative of individual component phenotypes present in a general population; further research is needed to characterize the pleiotropic effect of alleles in syndromic genes in persons without the syndromic disease.

## Introduction

Genetic syndromes are rare diseases defined by a specific and clinically recognizable set of phenotypes across multiple organ systems. The era of next-generation sequencing has enabled substantial progress in linking syndromic disease to specific genetic loci, coupled with public databases of genotype-phenotype relationships to facilitate the classification of genetic variants from "benign" to "pathogenic" for use in clinical decision making. Large population-scale databases of genetic variation without phenotypes, such as ExAC, have provided additional context for characterizing genotype-phenotype relationships in genetic disease [1]. For mutations previously thought to cause disease, population databases have often suggested lower estimates of penetrance than initially recognized [2, 3].

The diagnosis or classification of an individual with genetic syndrome relies upon expert recognition of a characteristic pattern of signs or symptoms or a set of defined diagnostic criteria. However, individuals displaying single phenotypes of a complex syndrome may not meet criteria for clinical diagnosis or genetic testing; expanding a binary definition of syndromic phenotypes to phenotype scores can identify more individuals with Mendelian disease patterns [4]. Similarly, individuals with clearly pathogenic mutations may be affected with only a single component phenotype of a genetic syndrome [5, 6]. However, the descriptions of allelic heterogeneity, penetrance, and expressivity in syndromic disease genes have focused almost exclusively upon rare or familial alleles [7, 8]

Recent studies have shown that rare and common variants in or near mendelian diseases genes are associated with complex traits in the general population [9–11]. Moreover, Freund et al [11] demonstrated an enrichment of signal from the summary statistics of Genome Wide

Association Studies (GWAS) near syndromic disease genes for the related phenotypes. However, this work was based on the curation of available GWAS summary statistics.

Here, we present a phenome-wide association study (PheWAS) approach to systematically explore expressivity of common and rare alleles in genes associated with four well-described syndromic diseases in the general population. Using the UK Biobank, we linked individual-level medical and morphometric data to the characteristic phenotypes of Alagille (AS), Marfan (MS), DiGeorge (DS), and Noonan (NS) syndromes. These data allow a survey of the association of common and rare alleles to single component phenotypes of each syndrome within the general (non-syndromic) population.

## Results

Based on the Human Phenotype Ontology (HPO)–an ontology-based system developed using medical literature and other ontology-based systems [12]–we identified 196 HPO terms related to AS, MF, DS, and NS. Of these 196 HPO terms, 53 were shared between at least two syndromes, and seven terms were included in all four syndromes (S1 Table). After grouping the HPO terms into categories based on affected organ systems, there were 115 HPO terms of which 73 could be matched to 100 phenotypes available in the UKBB. We additionally included liver and renal serum biomarkers such as alanine aminotransferase, creatinine, and direct bilirubin to capture liver and renal dysfunction observed in some of the genetic syndrome. Most of the unmatched phenotypes were related to specific abnormalities of body structure or the musculoskeletal system, which were poorly represented in clinical and billing codes, or measurements such as impaired T-cell function, not available in the UKBB.

### Characteristics of the study population

A total of 337,198 unrelated individuals were included in our analysis; the mean age was 65.8 years (sd = 8.0) and 53.7% of subjects were male. The number of subjects by phenotype is reported in S2 Table. Hypercholesterolemia (HP0003124), gastroesophageal reflux (HP0002020), premature osteoarthritis (HP0003088), and hypertriglyceridemia (HP0002155) were the most prevalent phenotypes with 12.8% (43,054 cases), 9% (30,229 cases), 8.9% (2,994 cases), and 8.6% (29,137 cases), respectively.

### Genotype-phenotype associations are common across all syndromic genes

We tested the associations between all variants and all phenotypes included in our study. A total of 1,824,564 tests (84 phenotypes x 21,721 variants) were performed. Overall, we found significant association between 20 phenotypes, and multiple variants across *JAG1*, *FBN1*, *PTPN11*, *SOS2*, *RIT1*, *RAF1*, *KAT6B*, *RASA2*, *MAP2K1*, *CBL*, *DGCR2 and COMT* (Fig 1A and S3 Table). Using stepwise conditional analysis implemented in GCTA, we identified a subset of 46 variants independently associated with those phenotypes (Fig 1B and S4 Table); among which 9 variants were associated with more 2 or more phenotypes (Fig 1B and Table 1). Among the phenotypes with significant associations, hypothyroidism (HP0000821), diastolic BP (HP0005117), systolic BP (HP0004421), standing/sitting height ratio (abnormality of body height; HP0000002), birth weight (small for gestational age; HP0001518), amount of subcutaneous adipose tissues or body fat percent (reduced subcutaneous adipose tissue; HP0003758), growth abnormality (HP0001507), body mass index (abnormality of body mass; HP0045081), hyperlipidemia (HP0003124), direct bilirubin level, creatinine level, aspartate amino transferase level, and alkaline phosphate phosphatase level were significantly associated with variants across multiples genes (Fig 1A and Fig 1B).

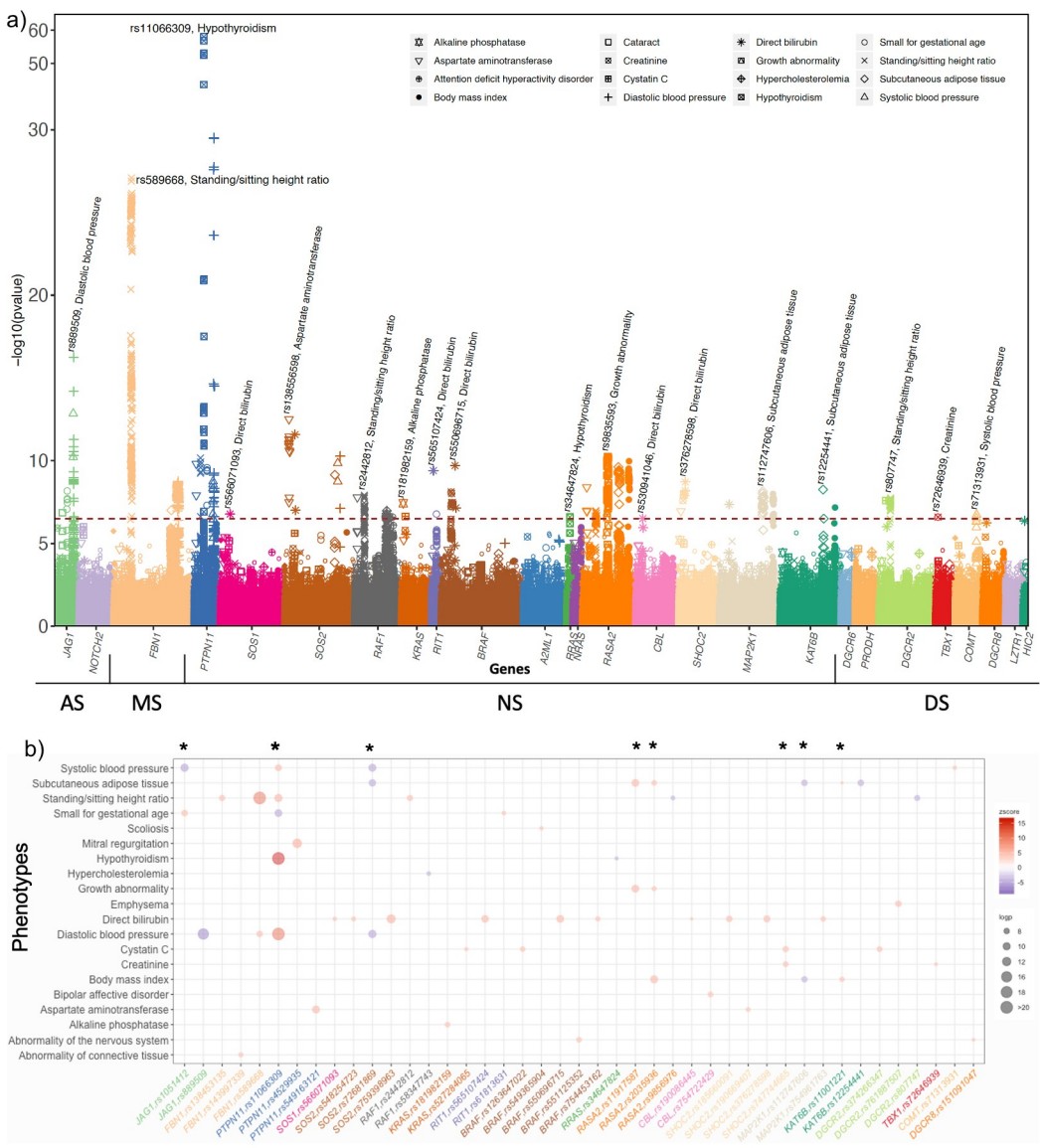

**Fig 1. Primary PheWAS results:** Variant level findings are displayed in panel (a) The red line represents the level of significance after Bonferroni correction (p<2.7x10$^{-07}$). The color indicates variant in each gene and the shape indicates each phenotype (ex: variants in *PTPN11* are represented in blue, and the association with diastolic blood pressure indicated with the sign +). Correlation plot of association between phenotypes and the subset of independent variants within each gene is displayed in panel (b). The points represent the z-score and the color represent the direction of the association. The color varies from purple (inverse association) to red (positive association). The size of the point corresponds to the p-value (-log10(p)); the stars indicate variants associated with multiple phenotypes.

The set of independent variants associated with multiple phenotypes are represented in Table 1. Diastolic BP and systolic BP along with body mass index displayed a genetic association in each of the four syndromes. Birth weight, subcutaneous adipose tissue (body fat percent), and tall stature for MS or short stature for NS are common phenotypes; while hypothyroidism, and growth retardation are reported in both NS and DS. In order to replicate our findings, we looked in the GWAS catalogue, for sets of variants in genes in association with phenotypes or proxy-phenotypes included in our study. We excluded studies performed

**Table 1. Table of association between the top SNPs with multiple phenotypes in each gene and significant (p-value < 2.7x10-07) or suggestive (p-value < 1x10-04) associations with HPO phenotypes (A1 corresponds to the effect allele).**

| RSID | HPO terms | Phenotype name | Genes | Ref Allele | Effect allele (A1) | Freq A1 | A1 count cases | A1 count controls | N | BETA | SE | P |
|---|---|---|---|---|---|---|---|---|---|---|---|---|
| **Marfan Syndrome** | | | | | | | | | | | | |
| rs589668 | HP0000002 | Standing/sitting height ratio | FBN1 | C | T | 0.25 | - | - | 334708 | 4.33E-04 | 3.96E-05 | 8.30E-28 |
| rs589668 | HP0005117 | Diastolic blood pressure | FBN1 | C | T | 0.25 | - | - | 329504 | 0.162 | 0.03 | 8.44E-09 |
| rs589668 | HP0003758 | Subcutaneous adipose tissue | FBN1 | C | T | 0.25 | - | - | 329504 | -0.076 | 0.02 | 2.22E-05 |
| **Alagille Syndrome** | | | | | | | | | | | | |
| rs1051412 | HP0004421 | Systolic blood pressure | JAG1 | A | C | 0.49 | - | - | 334475 | -0.279 | 0.04 | 6.08E-11 |
| rs1051412 | HP0001518 | Small for gestational age | JAG1 | A | C | 0.49 | - | - | 192000 | 0.012 | 0.002 | 6.33E-09 |
| rs1051412 | HP0000518 | Cataract | JAG1 | A | C | 0.49 | 18584 | 311632 | 335473 | -0.039 | 0.01 | 2.85E-04 |
| rs1051412 | HP0005117 | Diastolic blood pressure | JAG1 | A | C | 0.49 | - | - | 334708 | 0.088 | 0.02 | 3.11E-04 |
| rs1051412 | HP0003758 | Subcutaneous adipose tissue | JAG1 | A | C | 0.49 | - | - | 329504 | -0.053 | 0.02 | 7.22E-04 |
| **Noonan/RASopathy syndrome** | | | | | | | | | | | | |
| rs11001221 | HP0045081 | Body mass index | KAT6B | A | G | 0.08 | - | - | 333050 | 0.109 | 0.02 | 6.38E-08 |
| rs11001221 | HP0003758 | Subcutaneous adipose tissue | KAT6B | A | G | 0.08 | - | - | 329504 | 0.144 | 0.03 | 3.16E-07 |
| rs112747606 | HP0003758 | Subcutaneous adipose tissue | MAP2K1 | C | T | 0.23 | - | - | 329504 | -0.108 | 0.02 | 7.06E-09 |
| rs112747606 | HP0045081 | Body mass index | MAP2K1 | C | T | 0.23 | - | - | 333050 | -0.076 | 0.01 | 1.01E-08 |
| rs11066309 | HP0000821 | Hypothyroidism | PTPN11 | G | A | 0.41 | 17599 | 255384 | 335473 | 0.172 | 0.01 | 8.03E-59 |
| rs11066309 | HP0005117 | Diastolic blood pressure | PTPN11 | G | A | 0.41 | - | - | 334708 | 0.282 | 0.02 | 3.37E-30 |
| rs11066309 | HP0000002 | Standing/sitting height ratio | PTPN11 | G | A | 0.41 | - | - | 332526 | 2.27E-04 | 3.48E-05 | 6.91E-11 |
| rs11066309 | HP0001518 | Small for gestational age | PTPN11 | G | A | 0.41 | - | - | 192000 | -0.014 | 0 | 2.59E-10 |
| rs11066309 | HP0004421 | Systolic blood pressure | PTPN11 | G | A | 0.41 | - | - | 334475 | 0.255 | 0.04 | 2.71E-09 |
| rs11066309 | HP0045081 | Body mass index | PTPN11 | G | A | 0.41 | - | - | 333050 | -0.055 | 0.01 | 1.24E-06 |
| rs11066309 | HP0001297 | Stroke | PTPN11 | G | A | 0.41 | 4774 | 268208 | 335473 | 0.076 | 0.02 | 8.42E-05 |
| rs11066309 | Cystatin C | Cystatin C | PTPN11 | G | A | 0.41 | - | - | 313882 | 0.002 | 4.38E-04 | 8.56E-05 |
| rs2035936 | HP0045081 | Body mass index | RASA2 | G | T | 0.06 | - | - | 333050 | 0.158 | 0.02 | 1.01E-10 |
| rs2035936 | HP0003758 | Subcutaneous adipose tissue | RASA2 | G | T | 0.06 | - | - | 329504 | 0.187 | 0.03 | 4.30E-08 |
| rs2035936 | HP0001507 | Growth abnormality | RASA2 | G | T | 0.06 | - | - | 326878 | 0.064 | 0.01 | 7.58E-08 |
| rs2035936 | HP0001513 | Obesity | RASA2 | G | T | 0.06 | 1068 | 36401 | 335473 | 0.12 | 0.03 | 2.32E-04 |

*(Continued)*

**Table 1.** (Continued)

| RSID | HPO terms | Phenotype name | Genes | Ref Allele | Effect allele (A1) | Freq A1 | A1 count cases | A1 count controls | N | BETA | SE | P |
|---|---|---|---|---|---|---|---|---|---|---|---|---|
| rs11917587 | HP0001507 | Growth abnormality | RASA2 | G | A | 0.43 | - | - | 326878 | 0.035 | 0.01 | 2.13E-10 |
| rs11917587 | HP0003758 | Subcutaneous adipose tissue | RASA2 | G | A | 0.43 | - | - | 329504 | 0.099 | 0.02 | 2.42E-10 |
| rs11917587 | HP0000002 | Standing/sitting height ratio | RASA2 | G | A | 0.43 | - | - | 332526 | -1.49E-04 | 3.46E-05 | 1.71E-05 |
| rs11917587 | HP0045081 | Body mass index | RASA2 | G | A | 0.43 | - | - | 333050 | 0.046 | 0.01 | 4.32E-05 |
| rs747744665 | Cystatin C | Cystatin C | SHOC2 | A | C | 4.53E-04 | - | - | 313882 | 0.07 | 0.01 | 1.16E-08 |
| rs747744665 | HP0012100 | Creatinine | SHOC2 | A | C | 4.53E-04 | - | - | 313757 | 7.106 | 1.28 | 2.55E-08 |
| rs747744665 | HP0031970 | Urea | SHOC2 | A | C | 4.53E-04 | - | - | 313722 | 0.38 | 0.1 | 9.84E-05 |
| rs72681869 | HP0005117 | Diastolic blood pressure | SOS2 | G | C | 0.01 | - | - | 334708 | -0.76 | 0.12 | 5.26E-11 |
| rs72681869 | HP0004421 | Systolic blood pressure | SOS2 | G | C | 0.01 | - | - | 334475 | -1.293 | 0.2 | 1.41E-10 |
| rs72681869 | HP0003758 | Subcutaneous adipose tissue | SOS2 | G | C | 0.01 | - | - | 329504 | -0.457 | 0.07 | 7.12E-10 |
| rs72681869 | HP0045081 | Body mass index | SOS2 | G | C | 0.01 | - | - | 333050 | -0.251 | 0.05 | 2.14E-06 |
| rs72681869 | HP0001518 | Small for gestational age | SOS2 | G | C | 0.01 | - | - | 192000 | 0.046 | 0.01 | 3.26E-06 |
| rs72681869 | HP0000023 | Inguinal hernia | SOS2 | G | C | 0.01 | 365 | 7034 | 335473 | 0.213 | 0.06 | 1.32E-04 |
| rs72681869 | HP0001507 | Growth abnormality | SOS2 | G | C | 0.01 | - | - | 326878 | -0.092 | 0.03 | 3.65E-04 |

in non-Europeans or in the UK Biobank. We then extracted for each remaining phenotype, the association between candidate variants (defined variants significantly associated with the same phenotype in our study or in high LD (R2>0.8) with the associated variant) and the corresponding phenotype. The association between variants in *FBN1*, *PTPN11*, *SOS2*, and *JAG1* and, diastolic and systolic BP, were reported in the GWAS catalogue with the direction of effect concordant with the observed effect in our study. Similarly, the association between variants in *PTPN11* and hypothyroidism as well as the association between *JAG1* and birth weight were reported (S5 Table).

At a gene level, when using SKAT combined with weighted CADD score, 24 phenotypes including standing/sitting height ratio, blood pressure (systolic and diastolic), amount of subcutaneous fat, hypothyroidism, and hypercholesterolemia were significantly associated with several genes after multiple testing correction (p.fdr<0.05) (Table 2, Fig 2A and Fig 2B).

## Variation in syndromic genes are associated with component phenotypes

Marfan syndrome (MS) is a primary disorder of connective tissue with diagnostic criteria centered around cardiovascular, musculoskeletal, and ocular phenotypes linked to a single gene *FBN1* which encodes an extracellular matrix protein. Several variants in *FBN1* were significantly associated with increased standing/sitting height ratio and an elevated diastolic BP. An increased risk of aortic dissection and a lower percent of body fat (two major phenotypes in

**Table 2. Table of association between sets of associated phenotype–gene pairs after FDR correction (gene level association performed using SKAT test with variants weighted by their CADD score).**

| HPO | Phenotypes | genes | N markers | | | P-value SKAT weight CADD | |
|---|---|---|---|---|---|---|---|
| | | | Totals | Common | Rare | p-value | p.fdr |
| **Marfan syndrome gene** | | | | | | | |
| HP0000002 | Standing/sitting height ratio | *FBN1* | 1962 | 969 | 993 | 7.16E-35 | 4.28E-32 |
| HP0005117 | Diastolic blood pressure | *FBN1* | 1962 | 968 | 994 | 1.94E-11 | 3.17E-09 |
| HP0003758 | Subcutaneous adipose tissue | *FBN1* | 1962 | 968 | 994 | 1.33E-06 | 1.19E-04 |
| HP0004421 | Systolic blood pressure | *FBN1* | 1962 | 968 | 994 | 2.41E-05 | 1.50E-03 |
| HP0002647 | Aortic dissection | *FBN1* | 1962 | 968 | 994 | 1.18E-04 | 5.43E-03 |
| HP0000541 | Retinal detachment | *FBN1* | 1962 | 968 | 994 | 1.55E-03 | 3.91E-02 |
| **Alagille syndrome genes** | | | | | | | |
| HP0005117 | Diastolic blood pressure | *JAG1* | 580 | 257 | 323 | 1.01E-17 | 4.51E-15 |
| HP0001518 | Small for gestational age | *JAG1* | 580 | 257 | 323 | 4.45E-13 | 1.11E-10 |
| HP0004421 | Systolic blood pressure | *JAG1* | 580 | 257 | 323 | 3.11E-11 | 4.65E-09 |
| HP0003758 | Subcutaneous adipose tissue | *JAG1* | 580 | 257 | 323 | 1.08E-04 | 5.12E-03 |
| HP0000518 | Cataract | *JAG1* | 580 | 257 | 323 | 1.93E-04 | 8.25E-03 |
| HP0003124 | Hypercholesterolemia | *JAG1* | 580 | 257 | 323 | 1.60E-03 | 3.95E-02 |
| **Noonan syndrome / *RAS*-opathy genes** | | | | | | | |
| HP0000821 | Hypothyroidism | *PTPN11* | 832 | 263 | 569 | 1.58E-77 | 2.83E-74 |
| HP0005117 | Diastolic blood pressure | *PTPN11* | 832 | 263 | 569 | 6.32E-46 | 5.66E-43 |
| HP0004421 | Systolic blood pressure | *PTPN11* | 832 | 263 | 569 | 4.48E-16 | 1.61E-13 |
| HP0000002 | Standing/sitting height ratio | *PTPN11* | 832 | 263 | 569 | 4.91E-13 | 1.11E-10 |
| HP0001518 | Small for gestational age | *PTPN11* | 829 | 265 | 564 | 4.45E-12 | 8.87E-10 |
| HP0045081 | Body mass index | *PTPN11* | 832 | 263 | 569 | 3.63E-07 | 3.43E-05 |
| HP0003758 | Subcutaneous adipose tissue | *PTPN11* | 832 | 262 | 570 | 3.45E-05 | 2.06E-03 |
| HP0003081 | Increased urinary potassium | *PTPN11* | 832 | 262 | 570 | 6.16E-04 | 1.90E-02 |
| HP0001537 | Umbilical hernia | *PTPN11* | 832 | 263 | 569 | 1.01E-03 | 2.85E-02 |
| HP0001081 | Cholelithiasis | *PTPN11* | 832 | 263 | 569 | 1.24E-03 | 3.34E-02 |
| HP0003758 | Subcutaneous adipose tissue | *CBL* | 1295 | 388 | 907 | 2.43E-05 | 1.50E-03 |
| HP0045081 | Body mass index | *CBL* | 1295 | 386 | 909 | 7.22E-04 | 2.16E-02 |
| HP0000646 | Amblyopia | *CBL* | 1295 | 386 | 909 | 1.61E-03 | 3.95E-02 |
| HP0045081 | Body mass index | *KAT6B* | 1923 | 696 | 1227 | 2.26E-04 | 9.22E-03 |
| HP0000002 | Standing/sitting height ratio | *KAT6B* | 1923 | 696 | 1227 | 1.98E-03 | 4.52E-02 |
| HP0003758 | Subcutaneous adipose tissue | *MAP2K1* | 1555 | 616 | 939 | 3.62E-05 | 2.10E-03 |
| HP0045081 | Body mass index | *MAP2K1* | 1555 | 616 | 939 | 6.47E-05 | 3.62E-03 |
| HP0045081 | Body mass index | *NRAS* | 246 | 84 | 162 | 4.96E-13 | 1.11E-10 |
| HP0012603 | Urine sodium concentration | *NRAS* | 246 | 84 | 162 | 2.47E-10 | 2.77E-08 |
| HP0003758 | Subcutaneous adipose tissue | *NRAS* | 246 | 84 | 162 | 1.18E-08 | 1.24E-06 |
| HP0007018 | Attention deficit hyperactivity disorder | *NRAS* | 246 | 84 | 162 | 3.65E-04 | 1.30E-02 |
| HP0000002 | Standing/sitting height ratio | *NRAS* | 246 | 84 | 162 | 5.05E-04 | 1.68E-02 |
| HP0005117 | Diastolic blood pressure | *NRAS* | 246 | 84 | 162 | 7.66E-04 | 2.25E-02 |
| HP0000002 | Standing/sitting height ratio | *RAF1* | 1282 | 497 | 785 | 2.11E-10 | 2.61E-08 |
| HP0003124 | Hypercholesterolemia | *RAF1* | 1282 | 497 | 785 | 3.16E-06 | 2.70E-04 |
| HP0007018 | Attention deficit hyperactivity disorder | *RAF1* | 1282 | 497 | 785 | 8.70E-05 | 4.45E-03 |
| HP0001256 | Intellectual disability, mild | *RAF1* | 1281 | 493 | 788 | 3.43E-04 | 1.26E-02 |
| HP0003758 | Subcutaneous adipose tissue | *RAF1* | 1282 | 497 | 785 | 3.44E-04 | 1.26E-02 |
| HP0004421 | Systolic blood pressure | *RAF1* | 1282 | 497 | 785 | 5.94E-04 | 1.87E-02 |
| HP0001518 | Small for gestational age | *RAF1* | 1281 | 496 | 785 | 1.18E-03 | 3.26E-02 |

(*Continued*)

**Table 2.** (Continued)

| HPO | Phenotypes | genes | N markers | | | P-value SKAT weight CADD | |
|---|---|---|---|---|---|---|---|
| | | | Totals | Common | Rare | p-value | p.fdr |
| HP0003758 | Subcutaneous adipose tissue | *RASA2* | 1665 | 454 | 1211 | 1.13E-11 | 2.03E-09 |
| HP0001507 | Growth abnormality | *RASA2* | 1665 | 455 | 1210 | 2.18E-10 | 2.61E-08 |
| HP0000002 | Standing/sitting height ratio | *RASA2* | 1665 | 454 | 1211 | 2.96E-08 | 2.95E-06 |
| HP0045081 | Body mass index | *RASA2* | 1665 | 454 | 1211 | 1.27E-05 | 8.79E-04 |
| HP0001518 | Small for gestational age | *RASA2* | 1665 | 452 | 1213 | 9.16E-05 | 4.45E-03 |
| HP0012603 | Urine sodium concentration | *RASA2* | 1665 | 454 | 1211 | 3.18E-04 | 1.21E-02 |
| HP0001518 | Small for gestational age | *RIT1* | 234 | 90 | 144 | 2.15E-10 | 2.61E-08 |
| HP0001507 | Growth abnormality | *RIT1* | 234 | 90 | 144 | 4.25E-06 | 3.47E-04 |
| HP0045081 | Body mass index | *RIT1* | 234 | 90 | 144 | 5.98E-06 | 4.47E-04 |
| HP0003081 | Increased urinary potassium | *RIT1* | 234 | 90 | 144 | 7.94E-05 | 4.19E-03 |
| HP0003758 | Subcutaneous adipose tissue | *RIT1* | 234 | 90 | 144 | 3.16E-04 | 1.21E-02 |
| HP0000023 | Inguinal hernia | *RIT1* | 234 | 90 | 144 | 4.20E-04 | 1.45E-02 |
| HP0000120 | Creatinine clearance | *RIT1* | 234 | 90 | 144 | 1.33E-03 | 3.47E-02 |
| HP0004421 | Systolic blood pressure | *RRAS* | 171 | 67 | 104 | 5.71E-04 | 1.83E-02 |
| HP0001659 | Aortic regurgitation | *RRAS* | 171 | 67 | 104 | 1.87E-03 | 4.36E-02 |
| HP0001518 | Small for gestational age | *SHOC2* | 1183 | 421 | 762 | 9.18E-05 | 4.45E-03 |
| HP0001507 | Growth abnormality | *SHOC2* | 1183 | 423 | 760 | 2.41E-04 | 9.61E-03 |
| HP0001507 | Growth abnormality | *SOS1* | 1966 | 669 | 1297 | 1.27E-03 | 3.35E-02 |
| HP0005117 | Diastolic blood pressure | *SOS2* | 1792 | 731 | 1061 | 7.19E-04 | 2.16E-02 |
| **DiGeorge syndrome genes** | | | | | | | |
| HP0045081 | Body mass index | *COMT* | 691 | 275 | 416 | 5.62E-06 | 4.39E-04 |
| HP0004421 | Systolic blood pressure | *COMT* | 691 | 275 | 416 | 7.05E-06 | 5.06E-04 |
| HP0002155 | Hypertriglyceridemia | *COMT* | 691 | 275 | 416 | 1.45E-04 | 6.49E-03 |
| HP0003124 | Hypercholesterolemia | *COMT* | 691 | 275 | 416 | 2.24E-04 | 9.22E-03 |
| HP0003302 | Spondylolisthesis | *COMT* | 691 | 275 | 416 | 4.73E-04 | 1.60E-02 |
| HP0007018 | Attention deficit hyperactivity disorder | *DGCR2* | 1267 | 628 | 639 | 7.50E-05 | 4.08E-03 |
| HP0012603 | Urine sodium concentration | *DGCR2* | 1267 | 628 | 639 | 3.70E-04 | 1.30E-02 |
| HP0045081 | Body mass index | *DGCR8* | 603 | 231 | 372 | 1.93E-05 | 1.28E-03 |
| HP0004421 | Systolic blood pressure | *DGCR8* | 603 | 231 | 372 | 5.63E-04 | 1.83E-02 |
| HP0000002 | Standing/sitting height ratio | *DGCR8* | 603 | 231 | 372 | 1.51E-03 | 3.87E-02 |

MS) were observed for several of these variants although the association was merely suggestive (Fig 3A and S3 Table). All variants in *FBN1* displaying associations were located within the same LD block and were highly correlated with each other (Fig 3A and 3B). Using conditional regression analysis implemented in gcta on each phenotype, one independent signal was identified and rs589668 was tested with multiple phenotypes in *FBN1*. The variant rs589668 displays the top signal with high standing/sitting height ratio (p = $8 \times 10^{-28}$, Table 1 and Fig 1B), an elevated diastolic BP (beta = 0.02, se = 0.002, p = $8 \times 10^{-09}$), and a lower percent of body fat (beta = -0.08, se = 0,02, p = $5 \times 10^{-05}$, Table 1). The association observed with these phenotypes were as expected, in the same direction of effect as observed in Marfan syndrome. For instance, individuals with Marfan syndrome often have an elevated standing and sitting height ratio, and thin skin due to very small amounts of subcutaneous fat.

At the gene level association using the SKAT test with variants weighted by CADD score (Combined Annotation Depletion Dependent), standing/sitting height ratio, systolic and diastolic BP, subcutaneous adipose tissue, and aortic dissection were significantly associated with *FBN1* (Fig 2B and Table 2).

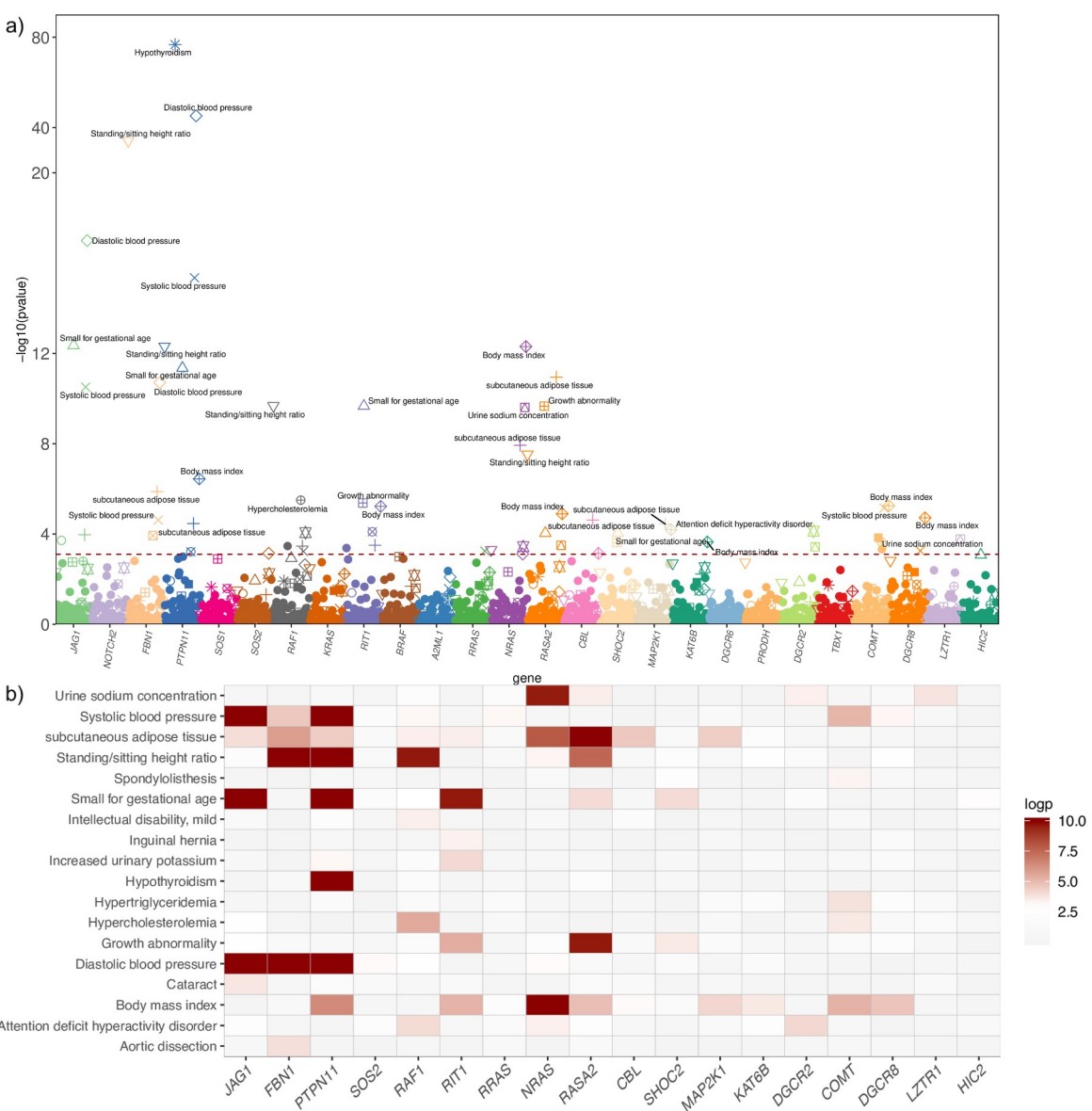

**Fig 2. Primary PheWAS results at a gene level.** Plot of PheWAS results for all genes and phenotypes (a). The red line represents the level of significance after FDR correction. Genes are represented by color and phenotypes are indicated by shape. Correlation plot for the set of significant gene–phenotype pairs. (b). The color represents the p-value of association and varies from none associated (grey-white) to significant association (red-dark red).

For NS and RAS-opathy related phenotypes, variants in *PTPN11* were associated with increased risk of hypothyroidism, high diastolic and systolic BP, and high standing/sitting height ratio (Fig 1A, Fig 1B and S3 Table). *variants* in *SOS2* were associated with lower systolic and diastolic BP, and lower percent of body fat (Fig 1A and S3 Table). *variants* in *MAP2K1*, and *KAT8B* were associated with lower body mass index, as well as lower level of cutaneous adipocyte tissues (Fig 1A, Fig 1B and S3 Table). variants in *RASA2* were associated with body mass index, level of subcutaneous adipose tissues, a lower ratio of standing/sitting height, and growth abnormality. We also found a significant association between variants in *SHOC2* and high level of Cystatin C, Creatinine, and Urea which reflect kidney dysfunction (Fig 1A, Fig 1B

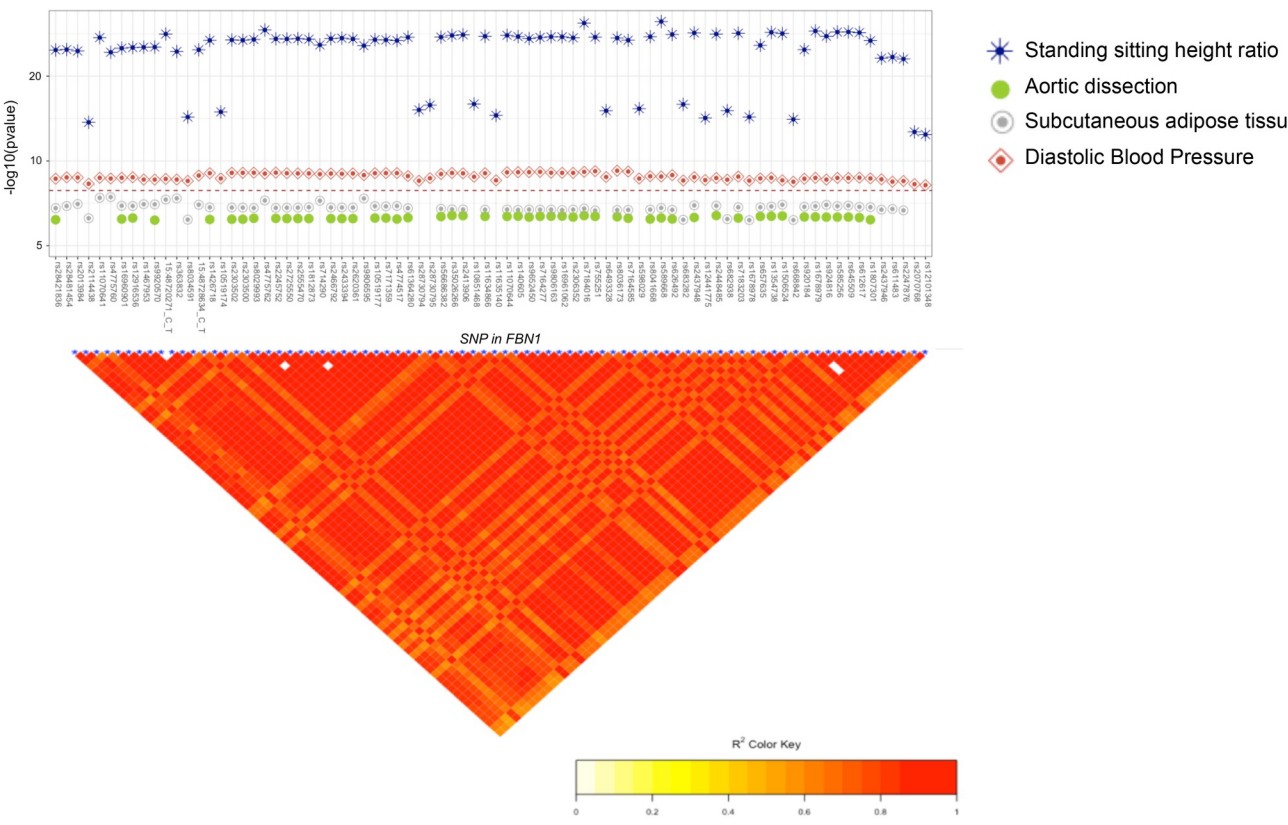

**Fig 3. PheWAS result and linkage plot for variants with pleiotropy in *FBN1*.** Associations between variants associate with multiple phenotypes (a) and linkage between the variants in *FBN1* (b). The red line represents the level of significance after Bonferroni correction (p = 2.7x10$^{-07}$).

and S3 Table). In addition, direct bilirubin concentration was significantly associated with variants in *SOS1*, *RIT1*, *CBL* and *SHOC2* (Fig 1A and S3 Table). variants in *PTPN11* display a moderate to low correlation, while high correlation was observed between variants in *RASA2*, *SOS2* and *MAP2K1* indicating that the association observed within each gene represents a single signal (Fig 4B). To identify independent signals within each gene for each associated phenotype, we performed conditional regression using stepwise selection procedure implemented in GCTA. For each subset of associated SNPs-phenotypes pairs within each gene, we identify one independent signal (Fig 1B). Among SNPs associated with multiple phenotypes, rs11066309 in *PTPN11* displays a strong association with increased risk for hypothyroidism (ALT freq = 0.40; OR [95% CI]: 1.19; [1.16–1.21]; p = 6x10$^{-59}$) along with five other phenotypes, including decreased body mass index (beta = -0.012, p = 1.13x10$^{-06}$) and birth weight (beta = -0.020, p = 2.95x10$^{-10}$) (Fig 1B and Table 1).

At a gene level, *PTPN11*, *NRAS*, *RASA2*, *SOS2*, *MAP2K1*, and *RAF1* were significantly associated with hypothyroidism, diastolic and systolic BP, birth weight, growth abnormality, subcutaneous adipose tissue, standing/sitting height ratio and body mass index (Fig 2B and Table 2).

Alagille syndrome is caused by mutations in *JAG1* and *NOTCH2* with major clinical manifestations in the heart and liver, and characteristic facial features. At a variant level, none of the AS specific phenotypes reached significance after multiple testing correction. However, suggestive associations were observed between and cataracts (p = 2.9x10$^{-04}$, S1(A) Fig). Several variants in *JAG1* were significantly associated with diastolic BP, systolic BP and birth weight

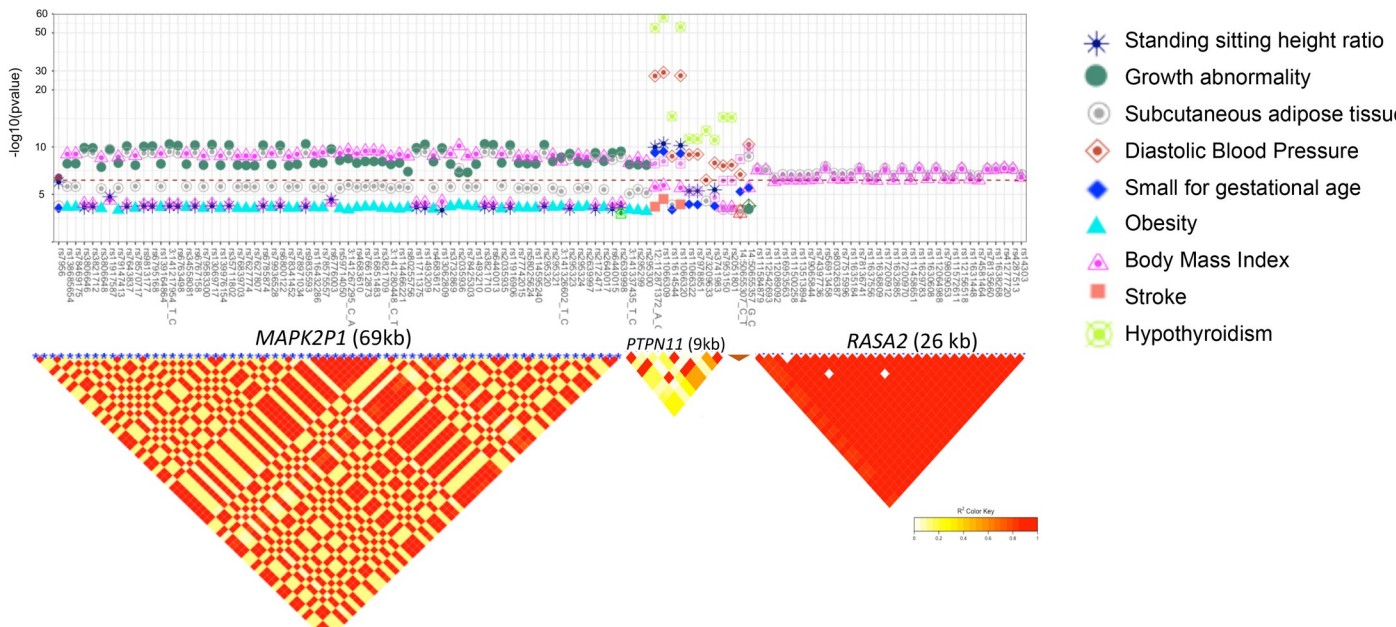

**Fig 4. PheWAS result and linkage plot for variants with pleiotropy in RAS-opathy genes.** Associations between variants with pleiotropic effects in *MAP2K1*, *PTPN11*, *SOS2* and *RASA2*, and HPO terms (a) and linkage between the variants (b). The red line represents the level of significance after Bonferroni correction (p = 2.7x10⁻⁰⁷).

(p<10⁻⁰⁸, Fig 1A, and S3 Table). The variant rs889509 displayed the most significant association with a lower diastolic BP (beta = -0.028; p = 8.2x10⁻¹¹, Table 1). At a gene level, *JAG1* was associated with diastolic BP (p = 3.48x10⁻¹⁵), birth weight (p = 3.84x10⁻¹⁰), systolic BP (p = 3.32 x10⁻⁰⁹) and cataracts (p = 2.21x10⁻⁰⁴, Fig 1B and S4 Table).

DiGeorge syndrome encompasses a recurrent microdeletion of multiple genes at the 22q11.2 locus due to the presence of segmental duplications, with affected individuals displaying neuropsychiatric, immunological, and cardiovascular phenotypes originating from defects in neural crest cell formation and migration. At the variant level, rs807747 in *DGCR2* and rs71313931 in *COMT* were significantly associated with abnormal body height (p = 2x10⁻⁰⁷) and systolic BP (p = 5.6x10⁻¹⁷), respectively (Fig 1A and 1B) while rs72646939 was associated with elevated creatinine levels (Fig 1A and 1B). At the gene level, *COMT* was associated with systolic BP, body mass index, and hypercholesterolemia while *DGCR2* was associated with attention deficit disorder, urine level concentration, and DGCR8 associated with body mass index, systolic blood pressure, and standing/sitting height ratio (p.fdr<0.05) (Table 2).

## Evaluation of pleiotropy and epistasis between phenotypes and genotypes

After identifying multiple pairs of variant-phenotype associations among which, 43 SNPs were independent; 9 of the 43 SNPs were associated with two or more phenotypes (Fig 1B). To access whether association between a single variant and multiple phenotypes are independent or due to correlation between phenotypes, we performed a formal test of pleiotropy between each independent variant and the associated phenotypes. We found a significant association between all pleiotropic SNPs and the associated phenotypes (S7 Table). Similarly, to evaluate whether associations between a phenotype and multiple variants were explained by variant-variant interaction or epistasis we performed a stepwise linear or logistic regression with interaction terms between pairs of associated variant for each phenotype. After multiple testing

correction, none of the variant pairs displayed significant interactions with each other (S8 Table).

## Discussion

Here, we systematically describe the association of variation in 26 mendelian genes linked to four syndromic diseases with the component phenotypes of the corresponding syndromic disease. We hypothesize that in the general population, common and rare alleles for syndromic diseases display pleiotropic effects with the phenotypes related to genetic syndromes. Using the UKBB, we linked individual-level data to the characteristic phenotypes of Alagille, Marfan, Noonan, and DiGeorge syndromes, showing clearly the association of common and rare alleles to single component phenotypes of each syndrome. Individual phenotypes that are modulated by variants within the loci syndromic disease genes appear to be present within the general population. These findings are consistent with the data reported by Bastarache et al [4] which suggested that scaling of component symptoms of rare disease into a continuous phenotyping score can improve the identification of individuals with rare diseases.

Within families of individuals affected by syndromic disease carrying the same pathogenic mutation, the expressivity of component phenotypes may vary in different individuals [13, 14]. Here, we show that many common and rare variants within loci of syndromic disease genes existing in the general population may result in expression of traits and phenotypes closely related to the syndrome of interest. For example, we observe associations of a common intronic variant in *FBN1* rs589668 (MAF = 0.25 in European populations) with increases in blood pressure and height and decreased subcutaneous fat distribution. In GTEx [15], this variant is an eQTL strongly associated with decreased expression in whole blood (p = $1.7 \times 10^{-37}$), which would be concordant with the known molecular mechanism of *FBN1* pathogenesis in MS: pathogenic alleles impairing gene function result in increased height and abnormal fat distribution and increased arterial stiffness [16, 17].

Modifiers of penetrance and phenotypic expressivity in Marfan syndrome have been previously proposed [18, 19] and a study based on a single Marfan syndrome family also suggested that differences in normal *FBN1* expression could contribute to the clinical variability of Marfan syndrome [19]. We observed four additional variants (rs11070641, rs4775760, rs363832 and rs140605) in *FBN1* to be associated with high standing/sitting height ratio, a characteristic feature often observed in Marfan syndrome. These variants are reported as benign variants in CLINVAR suggesting that they do not cause primary syndromic disease, but our data suggest they may be modifiers of penetrance for the phenotype of height. Our results suggest that common variants and local haplotype structure around syndromic genes may deserve more attention [20].

Noonan syndrome is caused by mutations in *PTPN11* and part of a group of related disorders arising from activating mutations in RAS-MAPK signaling pathway known as RASopathy which display many phenotypes across a variety of organ systems. A wide phenotypic variability and genetic heterogeneity have also been described in individuals with Noonan syndrome in relation to rare variants in *PTPN11* [22]. Here, we show that even in the general population, common and rare variants in *PTPN11* are independently associated with phenotypes such as hypothyroidism, small birth weight and low percent of body fat observed in some cases of Noonan syndrome [21–23]. In GTEx [15], numerous variants in *PTPN11*, such as rs11066309, rs3741983 and rs11066322 were significantly associated with a decreased expression in atrial appendage, adipose tissue, thyroid and skin and esophagus. Although in consistent with the role of *PTPN11* in thyroid function, cancer and autoimmunity [24–26], these

variants are instead described as eQTLs with *TMEM116*, *ALDH2* and *MAPKAPK5-AS1* located up to 500kb upstream of *PTPN11*, suggesting that the associations observed with rs11066309, rs3741983 and rs11066322 could potentially also arise from associations with these other genes.

Growth retardation, lower BMI and short stature are additional well-known characteristics of Noonan syndrome and display a phenotype-genotype variability of growth patterns in affected individuals [27]. In concordance with this study, we showed that in the general population, common and rare variants in *RASA2*, *SOS2* and *MAP2K1* are independently associated with growth characteristics (body mass index, height and growth abnormality) and the association driven by one or more haplotypes in each gene. Among *the* RAS-MAPK signaling pathway genes tested, we observe a significant association between the related phenotypes (lower body mass index, growth retardation, low percent of body fat) and rs3741983 (*PTPN11*), rs72681869 (*SOS2*), rs61755579 (*SOS2*), and rs112542693 (*MAP2K2*) reported in CLINVAR as "benign". Although these variants are indeed not sufficient to cause mendelian disease, they may nonetheless contribute to specific phenotypes related to Noonan syndrome when a "pathogenic" variant is present.

When performing genetic testing, allele frequency is often incorporated into an assessment of the pathogenicity of a genetic variant. Common variation in and around *JAG1* has previously been associated with such disparate phenotypes as pulse pressure, circulating blood indices, and birthweight, and none of the variants included in our analysis appeared to be directly associated with the component phenotypes of AS. However, the unifying molecular abnormality in AS are defects in vascular formation which lead to each of the component cardiovascular and liver phenotypes of Alagille syndrome [28, 29]. The pleiotropic effect detected for common alleles in *JAG1* (S7 Table) with multiple different phenotypes, may be linked to the underlying role in vascular formation.

Our study has some limitations. Our analysis was limited to phenotypes with more than 100 cases, and variants with minor allele frequency of at least 0.0001. Therefore, diseases with relatively rare prevalence or variants with extremely rare frequencies were not analyzed. In addition, because our study cohort consist of adults from the general population, specific phenotypes targeting facial and skeletal dysmorphism, such as butterfly vertebrae or broad forehead; specific abnormalities of organs, such as biliary disease were not present. However, to work around the absence of some phenotypes, we used proxy phenotypes or measurements present in the UKBB, such as head circumference as an alternative for broad forehead, education level for ADHD, weight and height at age 10 as proxy for growth abnormality. All things considered, the complexity of matching UKBB phenotypes to HPO terms may simply not capture some phenotypes, despite manual curation. An additional limitation of our study is the fact that, it is not possible to "diagnose" individuals from the data available in the UK Biobank in order to exclude them from analysis.

Key strengths of our study include the ability to systematically test multiple phenotype-genotype association and to highlight phenotypic expressivity of different variants linked to syndromic genes. Our study maps UKBB phenotypes to HPO terms and shows that common and rare variants in genes responsible for Alagille, Marfan, Noonan, and DiGeorge syndromes, are also independently associated with component phenotypes of these syndromes in the general population.

Our findings suggest that within the general population both common and rare variation in syndromic disease genes may be associated with component phenotypes of a syndrome. Further research on the expressivity of alleles in genes in the general population is needed to link our understanding of Mendelian syndromes with complex trait genetics.

## Materials and methods

### Study population and data collection

The study cohort was derived from the UK Biobank (UKBB), a large prospective cohort study with comprehensive health data from over 500,000 volunteer participants in the United Kingdom aged 37–73 years at recruitment in 2006–2010. The cohort has previously been described in detail [30–32]. Information on the UK biobank participants was collected at enrollment, and from electronic health record (EHR) information which includes diagnostic codes (ICD10, ICD9) and procedural codes (OPCS) from hospital admission records dating to 1992, and cancer registries. Data collected at the assessment visit included information on a participant's health and lifestyle, hearing and cognitive function, collected through a touchscreen questionnaire and verbal interview. A range of physical measurements was also performed, including blood pressure; arterial stiffness; body composition measures (including impedance); hand-grip strength; ultrasound bone densitometry; spirometry; and an exercise/fitness test with ECG. Samples of blood, urine, and saliva were also collected. Medical phenotypes were aggregated as previously described, incorporating available information including a broad set of medical phenotypes defined using computational matching and manual curation of on hospital in-patient record data (ICD10 and ICD9 codes), self-reported verbal questionnaire data, and cancer and death registry data [33, 34].

### Phenotypes of target syndromes

We identified phenotypes related to syndromic diseases through the Human Phenotype Ontology (HPO). HPO is an ontology-based system developed using medical literature, and other ontology-based systems such as Orphanet, and OMIM [12]. HPO provides a standardized vocabulary of phenotypic and abnormalities encountered in human diseases. The HPO has link symptoms/phenotypes to diseases or genetic disorders, and the causing genes. As an example, Alagille syndrome (AS) is linked to *JAG1*, and *NOTCH2* genes as well as all the phenotypes or symptoms observed in AS, such as atrial septal defect, hypertelorism, and butterfly vertebra.

HPO terms were directly matched to UKBB phenotypes when phenotypes in both systems had similar terminology. The direct phenotype matching was conducted using a semi-automatic mapping system which combines semantic and lexical similarity between word [35] followed by manual curation. When the HPO terms were not present, we performed an indirect matching by hand to find in the UKBB, the phenotype that best reflects the target HPO terms. For example, abnormality of body structure or body morphology such as abnormal body height, reduced subcutaneous adipose tissues, bone density or broad forehead, were respectively matched to sitting/standing height ratio; body fat percentage; bone mineral density, and head bone area. blood biomarkers measuring liver, and kidney functions such as direct bilirubin, creatinine, Alanine aminotransferase, Alkaline phosphatase, Gamma glutamyl-transferase were used as proxy for liver, or renal function. For psychiatric diseases such as depression and neurodevelopmental disorders such as attention deficit and hyperactivity disorder (ADHD), we used a score of depressive symptoms and self-reported educational level respectively as proxies for these terms.

To increase the number of subjects in some subgroup of phenotypes, we combined subcategories of HPO terms into a group or category. For example, 39 HPO terms representing an abnormality of head, ears, and eyes such as low-set ears, strabismus, macrotia, webbed neck, short neck, abnormality of the eye, microcornea, down-slanted palpebral fissure and other congenital abnormality of ears, were grouped into "Abnormality of head or neck

(HP0000152)" and mapped to icd10 targeting congenital malformations of eye, ear, face, and neck and other organs especially facial appearance (ICD10: Q10 to Q18 and Q87). Ten HPO terms for congenital abnormality of cardiovascular system including Ventricular septal defect, Atrial septal defect, Tetralogy of Fallot, Patent ductus arteriosus, Bicuspid aortic valve, Truncus arteriosus, Coarctation of aorta, Tricuspid valve prolapse were combined into abnormality of the cardiovascular system (HP0001626) and mapped to Congenital malformations of the circulatory system (ICD10: Q20 to Q28).

## Genotyping data

Genotyping was performed using the Affymetrix UK BiLEVE Axiom array on an initial 50,000 participants; the remaining 450,000 participants were genotyped using the Affymetrix UK Biobank Axiom® array. The two arrays are extremely similar (with over 95% common content). Quality control and imputation to over 90 million variants, indels and large structural variants was performed [35].

## Gene definitions

Using OMIM, and HPO, we identified 26 genes linked to the syndromes of our interest (Table 3). Each gene was defined from 5'UTR to 3'UTR with an extra additional 5kb upstream, and 5kb downstream the gene. To account for variants in regulatory elements of the target gene but located outside of the defined boundary, we additionally include within each target gene, variants located outside of the defined boundary but in eQTL in any tissue with the target gene. For the variant level association, we further extend the boundary of each gene to 50 kb upstream, and 50 kb downstream a gene. We identify a total of 21,712 variants in 26 genes related to Alagille syndrome, Marfan syndrome, Noonan syndrome, and DiGeorge syndrome were selected for our study (Table 2). The selected variants had a MAF $\geq$ 0.0001 and an imputation measurement (R2) $\geq$ 0.6

## Statistical analysis

**SNP level.** We performed the association between all 84 identified phenotypes, and all 21,721 variants in all the syndromic diseases genes included in our analysis. For binary traits, logistic regression with adjustment on age, sex, batch, and the top 5 principal components were used. First, regression was used in a situation of unbalanced numbers of cases and controls, especially when the number of cases was very small (less than 200 cases). For continuous traits, we performed linear regression with adjustment on age, sex, batch, and the top 5 principal components. Our analysis was restricted to individuals of European descent, due to the relatively small number of individuals from other ethnic groups in the UKBB. Bonferroni correction based on the number of independent tests was used to correct on multiple testing.

**Table 3. Table summarizing number of genes, SNPs and phenotypes (HPO terms) for each syndrome included in our analysis.**

|  | Genes | MAF>0.005, R2<0.8 | HPO terms | matched HPO term |
|---|---|---|---|---|
| Alagille Syndrome | *JAG1, NOTCH2* | 302 | 61 | 36 |
| Marfan syndrome | *FBN1* | 518 | 64 | 42 |
| Noonan syndrome | *PTPN11, SOS1, RAF1, KRAS, RIT1, BRAF, A2ML1, RRAS, SOS2, NRAS, RASA2, CBL, SHOC2, MAP2K1, KAT6B* | 3062 | 58 | 24 |
| DiGeorge syndrome | *22q11.2 deletion (TBX1, DGCR2, DGCR8, DGCR6, COMT, PRODH, HIC2, LZTR1)* | 1228 | 66 | 35 |

Given the high correlation between variants within gene or regions, Bonferroni correction is often stringent when the number of tests considered is number of SNPs time the number of phenotypes. To take in account the correlation between variants, we estimate the number of independent variants in a block of 50 kb with a correlation > 0.8 using the pairwise pruning method implemented in PLINK which estimated 2166 independent variants within our target regions. We apply a threshold of $2.7x10^{-07} = 0.05/(2166x84)$ independent tests. In order to replicate our finding, we looked in the GWAS catalog [36], for sets of variants in genes in association with phenotypes or proxy-phenotypes included in our study. We excluded studies performed in non-Europeans or in the UK Biobank. We then extracted for each remaining phenotype, the association between candidate variants (defined variants significantly associated with the same phenotype in our study or in high LD (R2>0.8) with the associated variant) and the corresponding phenotype

**Identification of an independent variant-trait association set.**   To identify a subset of variants independently associated with each phenotype, we performed the stepwise model selection for identification of variants independently associated with a phenotype; implemented in GCTA [37].

**Pleiotropy and epistasis assessment.**   To assess whether variants associated with multiple phenotypes reflect a correlation between phenotypes or are independently associated with each phenotype, we performed a pleiotropy test between each variant and the set of associated phenotypes using pleio, a R package for pleiotropy assessment. The association between each variant and a group of phenotypes were considered significant if the p-value were less than $2.1x10^{-04}$ (p<0.05/233 significantly associated variants). Similarly, to test for interaction between variants within a single gene or across different genes, we performed an epistasis test which consist of testing for the interaction between each pairs of associated variants and the corresponding phenotypes. We used linear regression for continuous phenotype and logistic regression for binary phenotypes with adjustment for age, sex and the first 10 PCs. An interaction term was considered significant if the pvalue were less than $3.7x10^{-05}$ based on Bonferroni correction (0.05/1320 tests).

**Gene level.**   At a gene level, we performed a Sequence Kernel association test (SKAT) using a sequence kernel method as well as a burden test [38, 39]. We performed the SKAT test on rare and common variants as well as on rare variants only. To account for the contribution of rare variants and common variants, we use SKAT CommonRare methods in which, rare and common variants are partitioned into two groups to test for the association with the phenotypes; the results of association is then combined using combined multivariate collapsing [38]. A variant was considered rare if the Minor allele frequency was less or equal to 0.05 (MAF ≤0.05). To account for their possible functional relevance, each variant was weighted in the SKAT test by their CADD score (Combined Annotation Depletion Dependent) [39, 40]. Although Gene-based SKAT tests are relatively insensitive, for sensitivity analysis, we also performed SKAT using allelic frequency. Each gene was defined from 5'UTR to 3'UTR with an extra 5kb upstream, and 5kb downstream. To account for variants in regulatory elements of the target genes but located outside of the defined boundary, we additionally include within each target gene, variants in eQTL with the target gene in any tissue but located outside of the defined boundary. We used FDR to correct for multiple testing.

## Supporting information

**S1 Table. Overall HPO term present in Alagille, Noonan, Marfan and DiGeorge syndromes.**
(XLSX)

**S2 Table. Description binary and continuous phenotypes.**
(XLSX)

**S3 Table. Subset of variant-Phenotype pairs with significant association.**
(XLSX)

**S4 Table. variant—phenotype association for the subset of independent variants from GCTA-cojo.**
(XLSX)

**S5 Table. Association between candidate variants and phenotypes of interested in non UK Biobank studies reported in the GWAS catalogue.**
(XLSX)

**S6 Table. Gene level association with different SKAT test P.weight CADD (rare variant test each variants are weight by their CADD score); P.RaCo (SKAT with adaptive sum test of rare and common variants); P.Burden (SKAT burden test with rare and common variants aggregation); SKAT rare (rare variant test only).**
(XLSX)

**S7 Table. Results of pleiotropy test between the all significant variants and the corresponding associated phenotype.**
(XLSX)

**S8 Table. Results of epistasis test between the subset of variants independently associated with each phenotype.**
(XLSX)

**S9 Table. Association between candidate variants and all phenotype reported in the GWAS catalog.**
(XLSX)

**S1 Fig. Diagram of phenotype matching system between the UK-biobank (UKB) phenotypes and HPO terms.**
(TIF)

**S2 Fig. Association between variants in *FBN1* and all HPO terms at the SNP and gene levels.**
(TIF)

**S3 Fig. Association between variants in *PTPN11*, and *gene in RAS/MAKP2* and all HPO terms at the SNPs and gene levels.**
(TIF)

**S4 Fig. Association between variants in *NOTCH2*, and *JAG1* and all HPO terms at the SNPs and gene levels.**
(TIF)

**S5 Fig. Association between variants in *22q11 locus* and all HPO terms at the SNPs and gene levels.**
(TIF)

**S6 Fig. Forest plots showing association between Clinvar variants in *FBN1*, *JAG1*, *PTPN11*, *MAP2K1* and *SOS2* and, MF, AS, NS-related phenotypes, respectively.**
(TIF)

**S7 Fig. Correlation plot between minor allele frequency and absolute value of beta for the subset of significant variant.** The color indicates variant in each gene and the shape indicates each phenotype (ex: variants in PTPN11 are represented in blue, and the association with hypercholesterolemia with the sign +). (a) set of variants with MAF<0.01, (b) set of variant with MAF between 0.01 and 0.5 (c) full set of significant variants.
(TIF)

## Author Contributions

**Conceptualization:** Erik Ingelsson, James R. Priest.

**Data curation:** Matthew Aguirre, Stefan Gustafsson, Priyanka Saha, Praneetha Potiny, Melissa Haendel, James R. Priest.

**Investigation:** Catherine Tcheandjieu.

**Methodology:** Catherine Tcheandjieu, Melissa Haendel.

**Software:** Matthew Aguirre, Manuel A. Rivas.

**Supervision:** Catherine Tcheandjieu, James R. Priest.

**Visualization:** Catherine Tcheandjieu.

**Writing – original draft:** Catherine Tcheandjieu, James R. Priest.

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
