## [Decision Letter · Decision Letter 0]

2 Dec 2019

Dear Dr Priest,

Thank you very much for submitting your Research Article entitled 'A phenome-wide association study of four syndromic genes reveals pleiotropic effects of common and rare variants in the general population.' to PLOS Genetics. Your manuscript was fully evaluated at the editorial level and by three independent experts who served as peer reviewers for this manuscript. As noted in the comments below the reviewers appreciated the attention to an important problem, but raised some substantial concerns in the current manuscript. Based on the reviews, we will not be able to accept this version of the manuscript, but we would be willing to review again a much-revised version. We cannot, of course, promise publication at that time.

If you decide to revise the manuscript for further consideration at PLOS Genetics, please aim to resubmit within the next 60 days, unless it will take extra time to address the concerns of the reviewers, in which case we would appreciate an expected resubmission date by email to plosgenetics@plos.org.

[LINK]

We are sorry that we cannot be more positive about your manuscript at this stage. Please do not hesitate to contact us if you have any concerns or questions.

Yours sincerely,

Santhosh Girirajan

Associate Editor

PLOS Genetics

Scott Williams

Section Editor: Natural Variation

PLOS Genetics

Reviewer's Responses to Questions

**Comments to the Authors:**

Reviewer #1: The authors present a study on common polymorphisms linked to Mendelian disease genes and show that some GWAS hits are associated with phenotypic features associated with the Mendelian diseases. This finding is not entirely new, although the authors do not cite much of the previous literature, e.g., Translating Mendelian and complex inheritance of Alzheimer's disease genes for predicting unique personal genome variants (PMC3277633), The Human Phenotype Ontology: Semantic Unification of Common and Rare Disease (PMC4572507), amd Phenotype-Specific Enrichment of Mendelian Disorder Genes near GWAS Regions across 62 Complex Traits (PMID:30290150), and others.

One novelty in the current study is related to its use of UK Biobank data, and the phenotypic focus of the current work, which I believe goes beyond what I am familiar with in the literatire.

1. There is a major mistake in the way the authors are using ClinVar data. They state:

"Four additional SNPs (rs11070641, rs4775760, rs363832 and rs140605) that reach genome-wide significance with high standing/sitting height ratio and diastolic BP were correlated with several syndromic disease entities in CLINVAR including stiff skin syndrome, ectopia lentis, MASS syndrome, thoracic aortic aneurysm and aortic dissection (SupplementaryTable 3). However, rs11070641 (which is https://www.ncbi.nlm.nih.gov/clinvar/variation/316310/) is listed as ClinVar as benign. The Table at ClinVar is listing conditions that in

principle can be tested for FBN1 mutations (including say stiff skin syndrome). However, it is not saying that the variant is linked with these diseases! Stiff skin syndrome is caused by highly specific mutations in exon 37 of FBN1 that affect an integrin binding site (review the OMIM entry https://omim.org/entry/184900 for details). rs11070641 is a 3UTR variant.

The authors should review all associations that have been derived in this way. It would be useful to have the actual evidence for association with disease reflect the actual data and not a ClinVar assertion (which is second hand so to speak, and the data are often noisy as in this case).

2. SNPs in PTPN11 display a moderate to low correlation with each other suggesting several independent signals within the locus

=> Do the authors have a biological explanation for the observation that only the PTPN11 gene was predicted to have independent signals?

3. Modifiers of penetrance and phenotypic expressivity in Marfan syndrome have been proposed,[15,16]

but our results suggest that common variants and local haplotype structure around

syndromic genes may deserve more attention[17].

=> This has been discussed by several papers in the literature including

Hutchinson S, Furger A, Halliday D, Judge DP, Jefferson A, Dietz HC, Firth H,

Handford PA. Allelic variation in normal human FBN1 expression in a family with

Marfan syndrome: a potential modifier of phenotype? Hum Mol Genet. 2003 Sep

15;12(18):2269-76. Epub 2003 Jul 22. PubMed PMID: 12915484.

Minor

1. The Y axis in Figure 1 is nearly impossible to read without mangification. The figure should be redone so that it can be interpreted at the size of a normal journal figure.

I do not follow what the authors mean by "HPO terms are represented by shape"? Please add a legend!

2. line 175 typo: lower level of cutaneous adipocytes tissues => ower level of cutaneous adipocyte** tissues

3. Figure 3 legend typo: RAS-opathie genes => RAS-opathy genes

4. in FBN1 rs589668 (MAF=0.25 in Europeans populations) => in FBN1 rs589668 (MAF=0.25 in European** populations)

5. line 239 typo: GTex => The correct acronym is GTEx

Reviewer #2: attached

Reviewer #3: This is a very interesting paper describing a PheWAS based on genes important for Mendelian syndromes. I really like this idea of looking for associations in these genes with symptoms of these syndromes, whereby some alleles may be less pathogenic and thus do not lead to the full syndrome. The paper is very well written and I am very excited about this work.

That all said, I am left with some questions and wondering about some analysis choices. Also, there are perhaps some additional analyses that would really strengthen the conclusions. I will list these ideas/questions below in the order that I came across them in the paper rather than order of importance.

1. When you did the gene burden testing in SKAT, as mentioned in abstract, is this with rare variants in those genes only? Or rare and common combined? I think this distinction is important.

2. In the author summary, you use the abbreviation 'SD'. This is not defined. I assume it is 'syndromic disease', but I am not certain.

3. I wondered why there was not a table of the gene-based results in the main paper?

4.Minor point: RAS-opathy is spelled differently throughout. Sometimes RAS-opathie. Sometimes RAS-opathy. It should be consistent.

5. In terms of the results section, several questions came to mind after reading it. Did you look at any phenotypes that are not characteristic of these syndromes? You could think of them as additional exploration or also a negative control. Did you do a formal test of pleiotropy using a method like 'pleio'? Did you do any conditional analyses to determine if the suggested pleiotropy associations are independent or whether they are present because of the correlation between the traits?

6. I think that GTEx is misspelled throughout.

7. In methods, how did you bin variants for SKAT? Rare only? Rare + common? How did you define genes in terms of basepairs (up/downstream of transcription start/end?)?

8. Did you also look at burden tests like regression or wilcoxon? It is known that SKAT can have a high type I error rate, so it would be a good idea to be conservative with those findings.

9. Much like GWAS, analyses like these could be prone to false positives. It is possible to look for replication of these signals either through independent datasets or even cross-validation within this large dataset.

10. Finally, did you consider doing a test of epistasis in these analyses? Since there is a lot of evidence of epistasis in the different symptoms of Mendelian disease, and epistasis and pleiotropy often co-occur, it would be very cool to look to see if there is any evidence for nonlinear interactions between these genes, or variants in these genes.

**Have all data underlying the figures and results presented in the manuscript been provided?**

Reviewer #1: No: Data from UKBB cannot be provided and so I do not believe the authors can do so. They do not provide code/scripts that were used, which should be possible.

Reviewer #2: Yes

Reviewer #3: Yes

PLOS authors have the option to publish the peer review history of their article (what does this mean?). If published, this will include your full peer review and any attached files.

Reviewer #1: No

Reviewer #2: No

Reviewer #3: No

---

## [Decision Letter · Decision Letter 1]

7 Apr 2020

Dear Dr Priest,

Thank you very much for submitting your Research Article entitled 'A phenome-wide association study of four syndromic genes reveals phenotypic expressivity of common and rare variants within the general population' to PLOS Genetics. Your manuscript was fully evaluated at the editorial level and by independent peer reviewers. The reviewers appreciated the attention to an important topic but identified some minor aspects of the manuscript that should be addressed.

We therefore ask you to modify the manuscript according to the review recommendations. Your revisions should address the specific points made by each reviewer. Specifically, please make the small editorial changes requested by reviewers 2 and 3. And as requested by reviewer 1, we would like you to make your code publicly available, GitHub is an excellent repository for this, but any open access location is acceptable; just note in the paper where the code will be available.. Once these changes are made we will be happy to accept your manuscript for publication.

[LINK]

Yours sincerely,

Santhosh Girirajan

Associate Editor

PLOS Genetics

Scott Williams

Section Editor: Natural Variation

PLOS Genetics

Reviewer's Responses to Questions

**Comments to the Authors:**

Reviewer #1: The authors have responded well to all of my comments but one.

In my original review, I stated that

They do not provide code/scripts that were used, which should be possible.

=> The authors seem to have ignored this comment. They should share the scripts they used for analysis and provide sufficient documentation such that others can reproduce their results. Of course they cannot share the UKBB data but new users can be expected to download this themselves.

Reviewer #2: This is a revision that I think has come back improved and more clearly written. This work, which looks at component phenotypes associated with 26 Mendelian disease genes using UK Biobank. It is quite exciting to think about how Mendelian disease genes can be used to interpret common variation that regulates those genes and component phenotypes to a lesser degree. Overall, I am excited by the work which starts to consider how we can unify our knowledge bases between Mendelian syndromes and complex trait genetics to improve our understanding of genetic-influence on human traits.

I don’t have major comments and minor comments are below

Minor:

line 178: should use “weighted” not weight

Figure 2: first line: should say “Plot of PheWAS results for all genes and all phenotypes”

Line 395: The last sentence of the manuscript is a little unclear and would benefit from some simplification to get their message across

Reviewer #3: The authors did a very nice job revising the manuscript based on the reviewer comments. One remaining issue that I see is that they changed the title, which I think was the right thing to do, however I think they have a typo. It now says four syndromic genes. In this study, they did a PheWAS for more than 4 genes. They used gene sets from four syndromic diseases. I think they meant four diseases.

Also in the discussion, the authors mention 4 syndromic loci. The explored 26 genes from the 4 syndromes. So it is unclear to me why they keep saying 4 syndomic loci.

Finally, the authors should be careful about how they refer to SNPs. Sometimes they use "SNPs" and other times "SNPS".

**Have all data underlying the figures and results presented in the manuscript been provided?**

Reviewer #1: No: It is not possible to share the UKBB data

Reviewer #2: Yes

Reviewer #3: Yes

PLOS authors have the option to publish the peer review history of their article (what does this mean?). If published, this will include your full peer review and any attached files.

Reviewer #1: No

Reviewer #2: No

Reviewer #3: No

---

## [Editor Report · Decision Letter 2]

27 Apr 2020

Dear Dr Priest,

We are pleased to inform you that your manuscript entitled "A phenome-wide association study of 26 Mendelian genes reveals phenotypic expressivity of common and rare variants within the general population" has been editorially accepted for publication in PLOS Genetics. Congratulations!

Yours sincerely,

Santhosh Girirajan

Associate Editor

PLOS Genetics

Scott Williams

Section Editor: Natural Variation

PLOS Genetics

Comments from the reviewers (if applicable):

**Data Deposition**

http://datadryad.org/submit?journalID=pgenetics&manu=PGENETICS-D-19-01777R2

**Press Queries**

---

## [Editor Report · Acceptance letter]

29 Oct 2020

PGENETICS-D-19-01777R2 

A phenome-wide association study of 26 Mendelian genes reveals phenotypic expressivity of common and rare variants within the general population 

Dear Dr Priest, 

We are pleased to inform you that your manuscript entitled "A phenome-wide association study of 26 Mendelian genes reveals phenotypic expressivity of common and rare variants within the general population" has been formally accepted for publication in PLOS Genetics! Your manuscript is now with our production department and you will be notified of the publication date in due course.

With kind regards,

Matt Lyles

PLOS Genetics

On behalf of:
